# UWB Sensing for UAV and Human Comparative Movement Characterization

**DOI:** 10.3390/s23041956

**Published:** 2023-02-09

**Authors:** Angela Digulescu, Cristina Despina-Stoian, Florin Popescu, Denis Stanescu, Dragos Nastasiu, Dragos Sburlan

**Affiliations:** 1Department of Communications and Information Technology, “Ferdinand I” Military Technical Academy, 050141 Bucharest, Romania; 2Faculty of Mathematics and Informatics, Ovidius University of Constanta, 900527 Constanta, Romania

**Keywords:** LSS UAV, UWB sensing, phase diagram entropy, movement map

## Abstract

Nowadays, unmanned aerial vehicles/drones are involved in a continuously growing number of security incidents. Therefore, the research interest in drone versus human movement detection and characterization is justified by the fact that such devices represent a potential threat for indoor/office intrusion, while normally, a human presence is allowed after passing several security points. Our paper comparatively characterizes the movement of a drone and a human in an indoor environment. The movement map was obtained using advanced signal processing methods such as wavelet transform and the phase diagram concept, and applied to the signal acquired from UWB sensors.

## 1. Introduction

Unmanned aerial vehicles (UAV), also known as drones, can play an essential role in future society by delivering commodities, the live broadcasting of events, maintaining surveillance and security, or providing mobile hotspots for broadband wireless access. Although drones should only be used for the benefit of our society, there are entities that have used them to carry out physical and cyber-attacks on infrastructure, private or public property, and individuals [1,2,3]. The attackers seem particularly interested in the low, slow, and small UAV (LSS UAV) type since it is highly difficult to detect, which is suitable for offensive and intelligence, surveillance, and reconnaissance (ISR) applications [4,5,6].

Therefore, drone detection and movement characterization represent important issues in maintaining safety and privacy in public and restricted areas. The current drone detection solutions are based on visual [7,8], acoustic [9], radio frequency (RF) [10,11,12], optoelectronic [13], and multimodal sensors [14,15,16].

The most popular approaches involve visual, acoustic, or visual–acoustic hybrid sensing and process the data using typical machine learning [15,17,18] or deep learning [7,19,20] based algorithms. The performances achieved by these methods are significantly influenced by the environmental conditions. The noisy environments represent a challenge for acoustic sensing methods, while rainy weather and low ambient light degrade the performance of vision-based drone detection systems.

Compared to the detection solutions relying on acoustic and visual sensing, both active and passive RF drone detection methods ensure higher robustness, achieving good results regardless of the environmental conditions [10]. The passive approaches, exploiting the physical layer protocols used by drones, analyze the RF spectrum to identify the uplink or downlink transmissions that occur between drones and their controller based on the well-known features of these links [21,22,23,24]. The major drawback of passive RF methods is their limitation in detecting the UAVs that function in fully autonomous mode.

As the active RF techniques provide good results in the previously mentioned challenging scenarios, detection is possible even in situations involving autonomous UAVs and the achieved performances are not strongly affected by external conditions [10,21]. However, LSS UAV detection still represents a demanding challenge for RF active approaches because of the very small radar cross section that characterizes these drones [25,26,27].

Due to their high-precision and capability to sense even in non line-of-sight (NLoS), the UWB sensing systems seem be a strong alternative in this field [28,29]. Hence, [30,31] presented several signal processing approaches that can be used in collaboration with UWB sensors to detect and characterize moving targets. In [30], the authors provided a promising solution for the multiple LSS UAV localization using the gradient descent method, whereas [31] addressed the problem of identifying the respiratory movement of human targets placed on the ground.

The choice of the indoor LSS UAV versus human movement detection and characterization is motivated by the fact that LSS UAVs represent a potential threat in the case of indoor intrusion, being used for unauthorized monitoring or spying activities [1,2,3], while the human presence is usually authorized. Considering the accuracy detection and its capability to sense even in NLoS, the active RF technique of UWB technology represents a suitable solution for the LSS UAV–human movement discrimination in indoor configuration.

Our approach, which continues the research work presented in [32], aims to detect and provide a comparative movement characterization of a LSS UAV versus a human using two UWB sensors connected in the same network. The acquired signals are processed using non-parametric methods such as the wavelet transform and the phase diagram concept. The performance of the parametric methods used in [30,31] strongly relies on the choice of the input parameters, while the choice of the use of the methods proposed in this paper was based on the fact the both methods do not need any a priori information regarding the sensed environment, and only rely on the characteristics of the emitted signal, information that is available for active sensing applications. Moreover, in the case of drone movement characterization, in [32], several classical methods were used and the state-of-the-art methods performed worse than the methods proposed in this paper.

The rest of this paper is organized as follows. Section 2 summarizes the signal processing methods used for the LSS UAV–human movement detection and characterization. Then, the experimental setup along with the implementation details and capabilities are specified in Section 3. In Section 4, we present the results obtained with each method for each configuration. Additionally, we carried out a comparative analysis of the results obtained in the case of drone movement versus human movement. Finally, we present the conclusions and the future perspectives in Section 5.

## 2. Theoretical Aspects

This section presents the signal analysis methods used for the discrimination between the LSS UAV and human movement using a classical method, the wavelet transform, and a more recent method, the phase diagram concept.

### 2.1. Wavelet Transform

This method is a linear transformation based on a dictionary and it is able to highlight the time-scale characteristics, thus being widely used as an efficient tool for signal analysis. The dictionary is a mathematical function of zero mean [32]:(1)$$\int_{-\infty }^{\infty }\psi_{0}\left(t\right)dt=0$$

Then, an orthonormal base (the dictionary) is created starting from this wavelet function and from its dilated and delayed variants. In this process, the wavelet is dilated with the scale parameter s and translated with the parameter τ.
(2)$$\psi_{\tau ,s}\left(t\right)=\frac{1}{\sqrt{s}}\psi \left(\frac{t-\tau }{s}\right)$$

In our approach, we considered the difference between each two successive scan lines as the input parameter, $d_{i}(t)$:(3)$$d_{i}(t)=s_{i}\left(t\right)-s_{i+1}\left(t\right),i=1,2,\ldots ,399$$
where $s_{i}\left(t\right)$ is the $i^{th}$ scan line of the UWB sensor. A scan line represents the measurement of the UWB received signal for a single pulse repetition period.

The analysis of the received UWB signals was performed by the correlation with the wavelet function [32]:(4)$$W_{i}\left(\tau ,s\right)=\frac{1}{\sqrt{s}}\int_{-\infty }^{\infty }d_{i}\left(t\right)\cdot \psi^{*}\left(\frac{t-\tau }{s}\right)dt,\;i=1,2,\ldots ,399$$

The movement map is given by the following equation [32]:(5)$$MM_{w}=\left[\begin{array}{c} \sum_{s}W_{1}\left(\tau ,s\right)\\ \sum_{s}W_{2}\left(\tau ,s\right)\\ \vdots \\ \sum_{s}W_{399}\left(\tau ,s\right) \end{array}\right]$$

With this approach, one important aspect concerns the definition of an appropriate wavelet family so that it resembles the signal of interest as much as possible, $d_{i}(t)$. The more similar the used dictionary is to the analyzed signals, the better the results are, as the correlation (Equation (4)) has higher values. In our approach, the dictionary was obtained using the emitted signals of the sensors [32].

### 2.2. Phase Diagram

This non-parametric method highlights the evolution of a time series in a new representation space. The phase diagram is the space in which all the possible states of a system are represented, with each state corresponding to a unique point in the diagram.

Using the difference between each two successive scan lines, $d_{i}(t)$, the phase diagram is defined by [32,33,34,35,36,37]:(6)$$\vec{v_{j}}=\sum_{k=1}^{m}d_{i}\left[j+\left(k-1\right)d\right]\cdot \vec{e_{k}},\;i=1,2,\ldots ,399$$
where $\vec{e_{k}}$ are the axis vector units. The choice of the embedding dimension m and the time delay d can be found in [33].

Figure 1 shows the phase diagram representation. For an m-dimensional phase diagram trajectory, the information can be quantified through the computation of a pair-wise distance.

The time-distributed recurrence (TDR) emphasizes the sudden, abrupt changes, namely, the transient signals from the trajectory (in our case, the UWB signals) [32,33,34,35]:(7)$$TDR_{i}=\sum_{j=1}^{M}\left(\left\|\vec{v_{j}}-\vec{v_{k}}\right\|\right),\;i=1,2,\ldots ,399,\;\;\;\;\;j,k=\left\{1,2,3,\ldots ,M\right\}$$
where M=N−(m−1)d; N is the number of samples of the signal $d_{i}\left(t\right)$; and the symbol ‖·‖ represents the Euclidian distance.

Next, the movement map given by the TDR quantification is defined as [32]:(8)$$MM_{TDR}=\left[\begin{array}{c} TDR_{1}\\ TDR_{2}\\ \vdots \\ TDR_{399} \end{array}\right]$$

This method has the advantage in that it does not require any a priori information and it can be used for many types of signals [32,33,34,35,36,37].

### 2.3. Phase Diagram Entropy

This approach represents our newest quantification method of the phase diagram [38,39]. The spatial distribution of the phase diagram vectors was computed in order to highlight the different phenomena that occur in a system.

We first determined the number of points that were close within a σ radius distance from each phase diagram representation point [38,39]:(9)$$N_{C_{j}}\left(d,m,\sigma \right)=\sum_{j=1,j\neq i}^{N-(m-1)d}\Theta \left(\left\|\vec{v_{\left[i\right]}}-\vec{v_{\left[j\right]}}\right\|-\sigma \right)$$
where Θ is the Heaviside function and ‖·‖ is the operator of Euclidean distance.

The choice of the threshold σ is considered as the ratio between parameters of the ellipse in which the phase diagram trajectory is included, as shown in Figure 2, where *a* is the major semi-axis of the ellipse and *b* is the minor semi-axis of the ellipse.
(10)$$\sigma =\frac{ab}{\left(a+b\right)}$$

Then, the ratio of the number of close points to the total number of vectors was calculated as:(11)$$T\left(d,m,\sigma \right)=\frac{N_{C_{j}}\left(d,m,\sigma \right)}{N-(m-1)d}$$

Next, the number of total points that satisfy the mentioned criterion was determined and normalized to the number of all vectors in the phase diagram:(12)$$P\left(d,m,\sigma \right)=\frac{1}{N-(m-1)d}\sum_{i=1}^{N-(m-1)d}\log\left(T_{i}\left(d,m,\sigma \right)\right)$$

The phase diagram entropy (PDEn) was used to study the behavior of the system by measuring the changes that occur as an increase in the embedding dimension [38,39].
(13)PDEn(d,m,σ)=P(d,m,σ)−P(d,m+1,σ)

Next, the movement map is given by the following equation:(14)$$MM_{TDR}=\left[\begin{array}{c} PDEn_{1}\\ PDEn_{2}\\ \vdots \\ PDEn_{399} \end{array}\right]$$

Like the previous approach, the phase diagram entropy has the advantage that it does not require any a priori information about the analyzed signal.

## 3. Experimental Setup

The measurements were performed using two UWB sensors, PulsON 440 [40], shown in Figure 3a. Both sensors were connected to a laptop as presented in Figure 3b. The S1 and S2 sensors were placed at the same height. We chose to use two sensors because the human/drone movement was carried out in two planes: horizontal or vertical.

This method can be generalized for more than two sensors, 3–4 sensors, or even more. The more sensors are placed, the more accurate the movement can be characterized. The sensors will not interfere because they are placed in the same network using the ALOHA protocol in monostatic radar configuration. This means that each sensor transmits and receives at a time for a given period (related to the maximum distance to be ranged). In our experiments, the maximum range distance was 5 m, therefore, the transmission-reception time for each scan line of each sensor was approximately 24 ns. This value of the sensing time of each of the two sensors is equivalent to an instantaneous sensing relative to the human/drone movement. Increasing the number of UWB sensors also increases the total sensing time of the system, which should be taken into consideration in terms of the computation time and resources.

The measurements of the human movement and the LSS UAV movement were individually performed by keeping the framework setup and relative position of the target unchanged. Therefore, in the monostatic radar configuration, we discriminated between the reflections occurring from the human body and from a UAV. From the theoretical analysis, the reflections must have different characteristics for the two media (human and LSS UAV) because of at least the following two reasons:The radar cross section (RCS) of a target can be seen as a comparison of the strength of the reflected signal from a target to the reflected signal from a perfectly smooth sphere with a cross-sectional area of 1 m^2^ [41]. As a rule, the larger an object, the stronger its radar reflection and thus the greater its RCS.In the interaction of the electromagnetic wave with the human body, respectively, the drone conducts to absorption [42] or reflection mechanisms. The dielectric parameters of the two media (human body and drone) are different [43].

The higher the impedance mismatches, the higher the reflection. The impedance mismatch occurs at the interface between air/skin in the case of the human and air/plastic in the case of the drone. Both human skin and plastic materials can be considered as non-magnetic media, thus their relative magnetic permeability is very close to unity. The conductivity of human skin in the frequency range of the UWB sensors varies between 17 and 31 S/m according to [44] while it is relative permittivity ranges between 38 and 35. Plastic materials can be considered perfect dielectrics, so their conductivity can be considered null. In this case, the wave impedances are different for the two media, resulting in different reflection coefficients that influence the outcome of the radar measurement, namely, the received signals.

Each of the performed measurements, Figure 4, was carried out for approximately 40 s to obtain a total of 400 scan lines in two configurations: left–right and forward–back movement of the target. The total sensing time defines the total number of scan lines. The S1 sensor was considered as the reference. The sensors were set to have the highest transmit gain [40] in a mono-static radar configuration on both ports (A and B) where the standard Time Domain BroadSpec antennas [45] were connected.

The emitted UWB pulses had the following characteristics: the bandwidth of [3.1 GHz, 5.3 GHz] and the maximum transmit power spectral density of −41 dBm/MHz [40].

The used UAV was a Parrot Mambo FPV drone with the following characteristics: it has four rotors, a weight of 63 g, and the dimensions of 18 cm × 18 cm × 4 cm [46]. The control was carried out with a dedicated smartphone application via Bluetooth.

## 4. Results

This section presents the results obtained for each configuration (forward–back movement, respectively, left–right movement of the target relative to S1) using the methods described in Section 2. The representations of the movement maps are plotted in range (in meters) instead of time (in seconds).

### 4.1. Forward–Back Movement

In this experimental setup, the movement of the target was performed relative to sensor S1 in the range 1–2.8 m, at a height of approximately 1.8 m. Thus, for the forward–back movement of the targets relative to S1 (the reference for this experimental part), the movement relative to the sensor S2 was left–right and vice versa.

Figure 5, Figure 6 and Figure 7 show that the S1 sensor senses the forward–back movements in time as spike motions, while the S2 sensor perceives them as oscillatory motions. These results show that the TDR approach provides better results than the wavelet analysis because it significantly reduces the salt and pepper noise from the representation. However, the analysis based on the PDEn approach is better suited, because it provides a clearer representation of an improved version of the TDR approach. In the representation based on PDEn, the amount of existing noise was minimal.

The discrimination between the LSS UAV movement and the human movement was given by the amplitude of the UWB reflected impulses, which were higher for the human because of its larger number of reflection points. Moreover, the trajectory described on the movement map by the drone was not as ‘periodic’ as the movement map trajectory of the human.

### 4.2. Left–Right Movement

In this configuration, the movement of the targets was performed at a distance of 2.5 m from sensor S1, but in the direction of sensor S2 in a forward–back movement. Therefore, the movement relative to sensor S1 is a left–right movement. We mention that during the experiment, the movement of the LSS UAV was not purely left–right relative to sensor S1, instead, it was combined with a forward movement.

In Figure 8, Figure 9 and Figure 10, it can be observed that the S2 movement map describes a spike motion. For the LSS UAV case, the spikes reduced their amplitude, meaning that the drone distanced itself from the sensor, whereas for the human movement, the amplitude of the spikes was the same, meaning that the movement remained unchanged.

On the side of S1, for the LSS UAV movement, the motion was oscillatory and around the scan lines 150–200 and 350–399, the forward component of the movement is highlighted. The human movement trajectory at S1 is oscillatory, but blurry because of the human position (from the side).The characteristics specific to the discrimination of the targets remain the same as in the previous subsection.

Moreover, because of the small dimensions of the LSS UAV, the device presented an instability in the air, which was also highlighted by our proposed algorithm. In contrast, the human movement trajectory presented uniformity.

### 4.3. Discussion

The choice of the indoor LSS UAV and human comparative movement characterization is given by the fact that such devices represent a potential threat for indoor intrusion for unauthorized surveillance/spying purposes, while humans have to pass several security points in order to access a restricted area.

This discussion regards the discrimination between the sensed targets in the LSS UAV versus human movement using the proposed approaches.

Based on the values obtained for each movement map, we can see that the values of the movement map matrix varied from 0 to a maximum value. This value varied according to the target for the wavelet transform and the TDR approach while it remained constant for the PDEn approach. Moreover, for both configurations, on the obtained movement maps, it can be seen that the noise was highly reduced with the PDEn approach. This effect was quantified and presented in Figure 11 and Figure 12 with the histogram applied on the movement maps from Figure 5, Figure 6 and Figure 7 for S1.

With these characteristics, we can see that the PDEn approach outperformed the other proposed approaches. As it can be seen in Figure 13, with this method, we could discriminate between the targets: the LSS UAV movement map had few values above 0.007, while the human movement map presented a quasi-uniform distribution of the values in the range [0.007, 0.015].

The same characteristics of the histogram remained valid for the S2 sensor and for the left–right configuration.

## 5. Conclusions

This paper proposes an alternative solution for LSS UAV and human movement discrimination using advanced signal processing methods and the UWB sensing system.

The UWB sensing system has the advantage that this technology is able to detect small moving objects and it has the capability to sense in NLoS, overcoming the visual sensing system limitations, while it does not interfere with other existing communication technologies.

With the use of the wavelet transform, the TDR approach, and the PDEn method, the human movement was characterized by a uniform trajectory on the movement map, whereas the LSS UAV movement map had lower values, a similar trajectory as the human, but not as steady.

Based on the results obtained in [32], the wavelet transform approach was used to highlight the movement trajectory of both targets. With the appropriate choice of the mother wavelet function and the required computation time/resources, these targets can be discriminated. Still on the movement map, the noise is disturbing, and in scenarios with higher distances (than the distances considered in our experiment), the trajectory was harder to characterize.

The TDR method provides better results for the movement map trajectory, pointing it out and eliminating the salt-and-pepper noise. This quantification measure, TDR, has the advantage of highlighting the transient signals in the phase space and to minimize the noise effect. As mentioned in [32], these transient signals are the UWB received pulses.

The PDEn approach is a new quantification method of phase diagram representation and has the advantage that it does not depend on the values of the pair-wise distances from the phase diagram. Hereby, in this paper, it was shown that the movement maps best highlighted the movement trajectory. Moreover, with this approach, the noise was minimized compared to the other two applied methods.

Hereby, this work proposes a new approach for the LSS UAV–human discrimination in an indoor scenario using a UWB sensing system and analyzing the results with non-parametric signal processing methods. Additionally, we computed the histogram of the movement maps for the PDEn approach because of the movement map values’ invariance and noise minimization. As discussed, the histogram analysis provides a discrimination between the human and LSS UAV movement map based on the obtained distributions that clearly separate between targets.

With these observations, our future work will focus on applying our approach in multiple configurations and with different types of targets. The acquired signals will represent the input data for a machine learning classification algorithm.

## Figures and Tables

**Figure 1 sensors-23-01956-f001:**
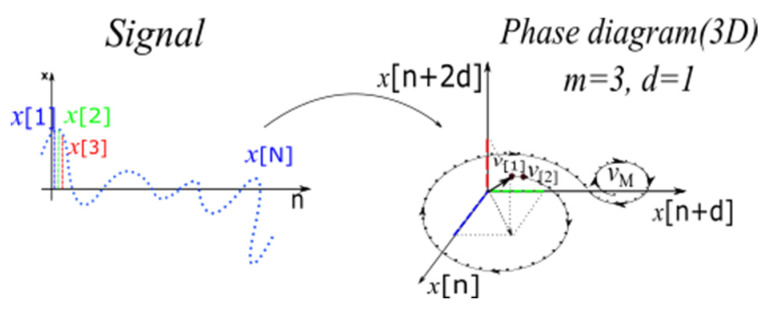
Illustration of the phase diagram concept.

**Figure 2 sensors-23-01956-f002:**
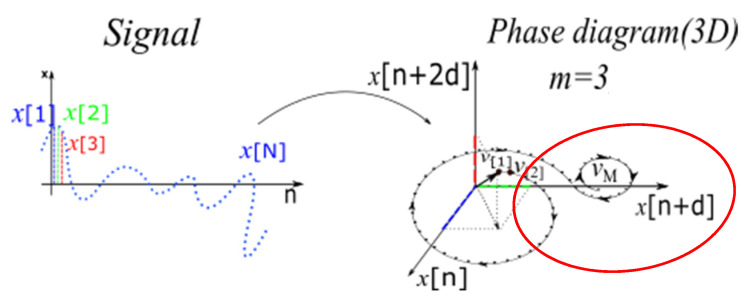
Inscription of the phase diagram representation in an ellipse (red) that defines the threshold σ.

**Figure 3 sensors-23-01956-f003:**
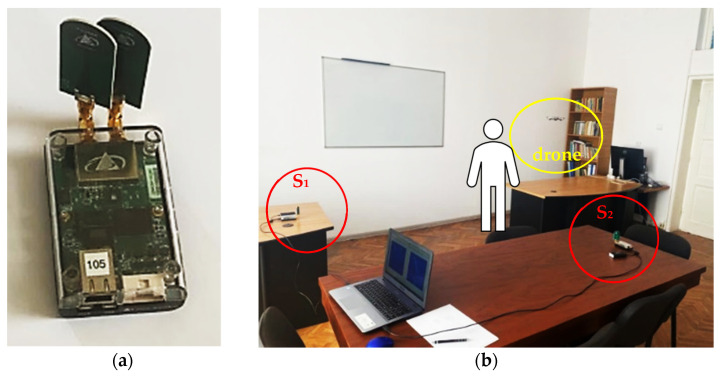
The PulsON 440 UWB sensor (**a**) used for the LSS UAV–human movement characterization and their placement (**b**) for the experimental setup.

**Figure 4 sensors-23-01956-f004:**
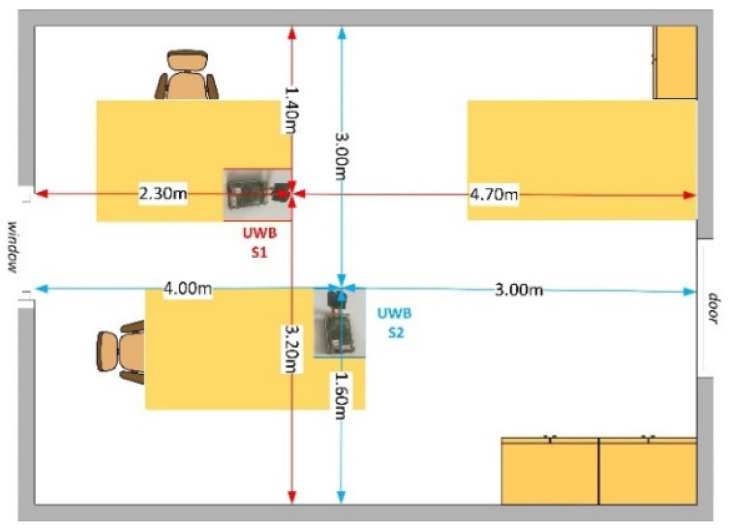
The room placement [32].

**Figure 5 sensors-23-01956-f005:**
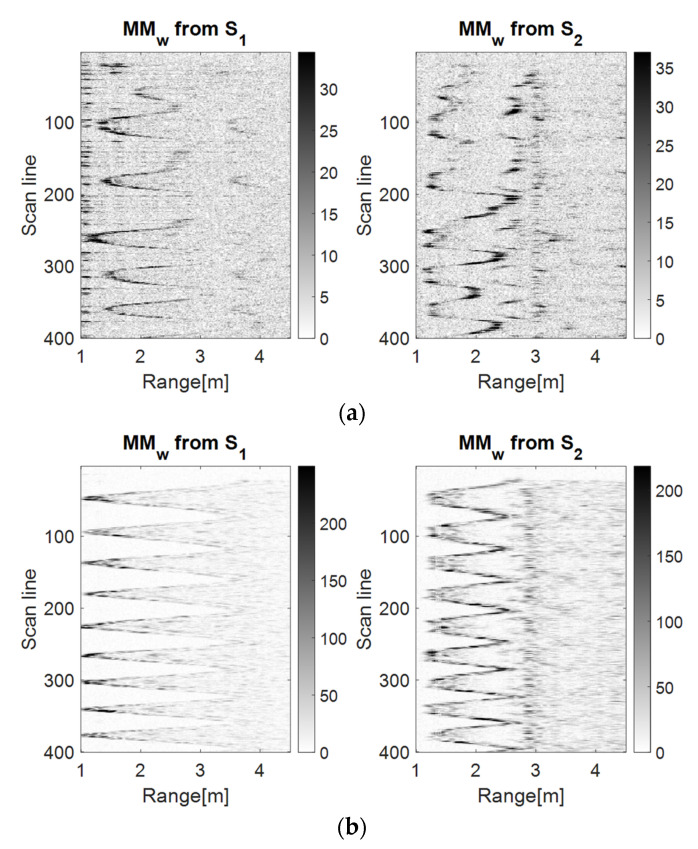
The drone (**a**) and the human (**b**) forward–back movement maps using the wavelet analysis using the Morlet wavelet as the mother function.

**Figure 6 sensors-23-01956-f006:**
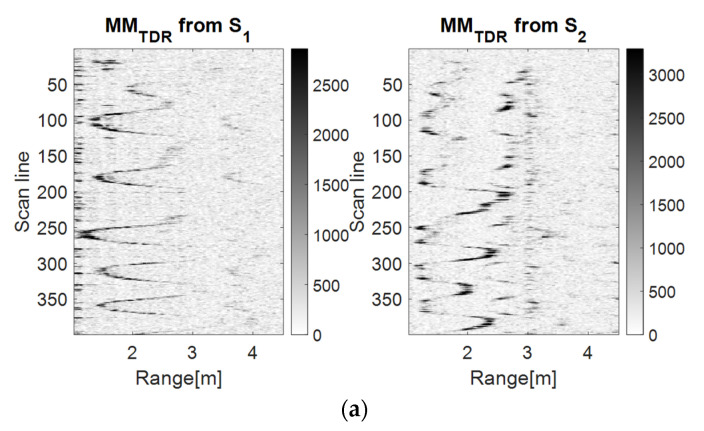
The drone (**a**) and the human (**b**) forward–back movement maps using the TDR method with *m* = 5, *d* = 1.

**Figure 7 sensors-23-01956-f007:**
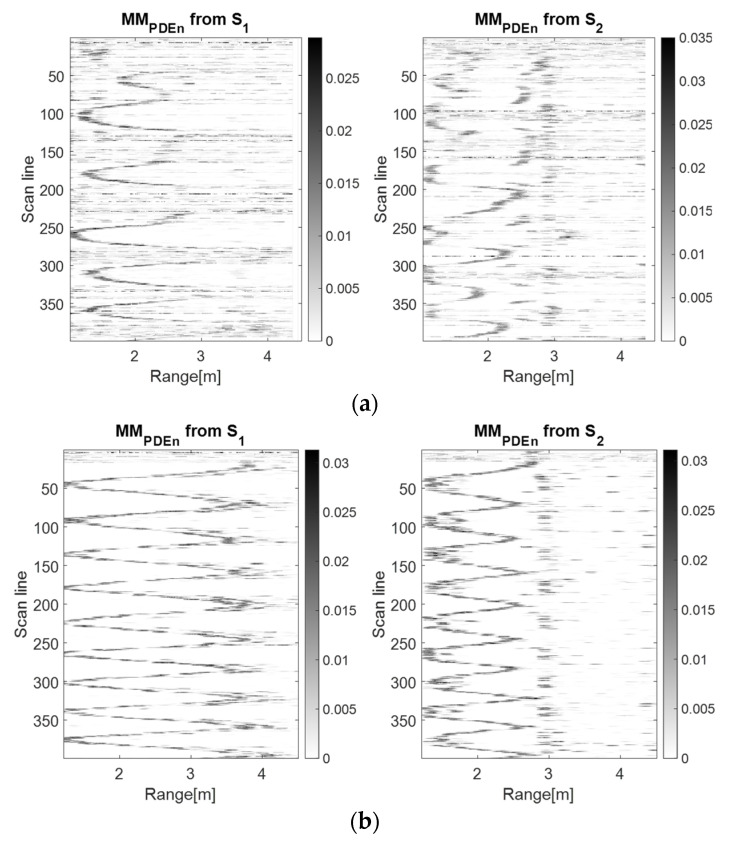
The drone (**a**) and the human (**b**) forward–back movement maps using the PDEn method with *m* = 5, *d* = 1, *σ* = 0.71.

**Figure 8 sensors-23-01956-f008:**
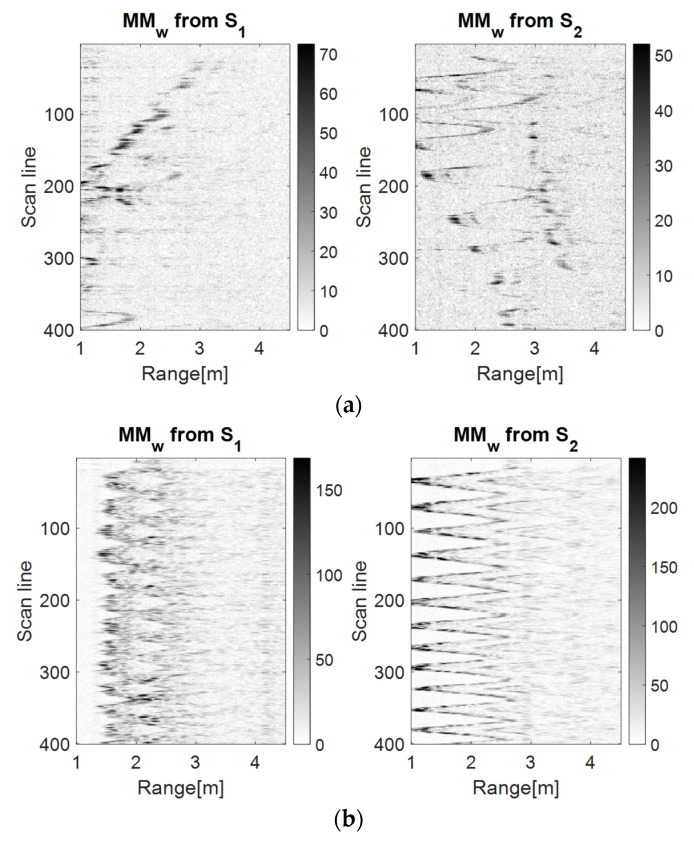
The drone (**a**) and the human (**b**) left–right movement maps using the wavelet analysis using the Morlet wavelet as the mother function.

**Figure 9 sensors-23-01956-f009:**
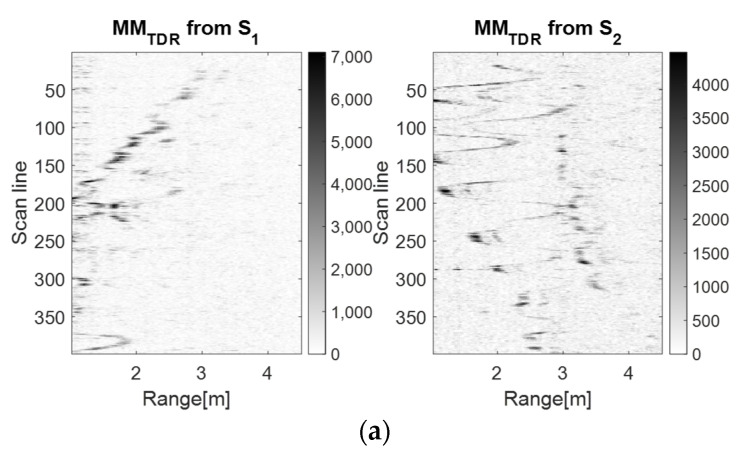
The drone (**a**) and the human (**b**) left–right movement maps using the TDR RPA method with *m* = 5, *d* = 1.

**Figure 10 sensors-23-01956-f010:**
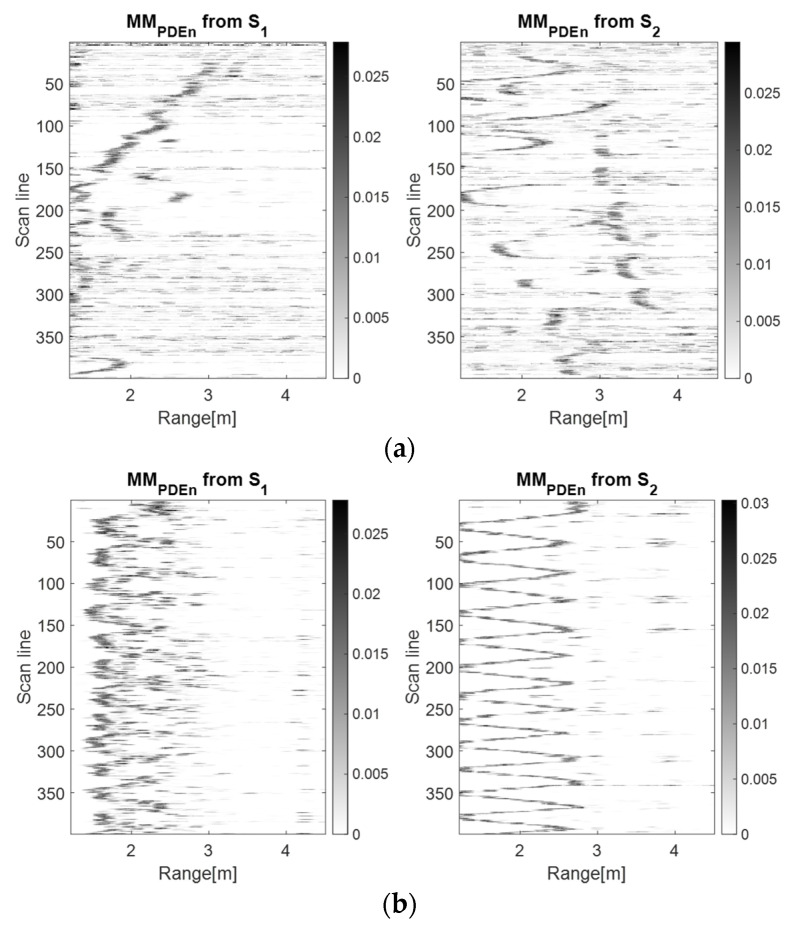
The drone (**a**) and the human (**b**) left–right movement maps using the PDEn method with *m* = 5, *d* = 1, *σ* = 0.71.

**Figure 11 sensors-23-01956-f011:**
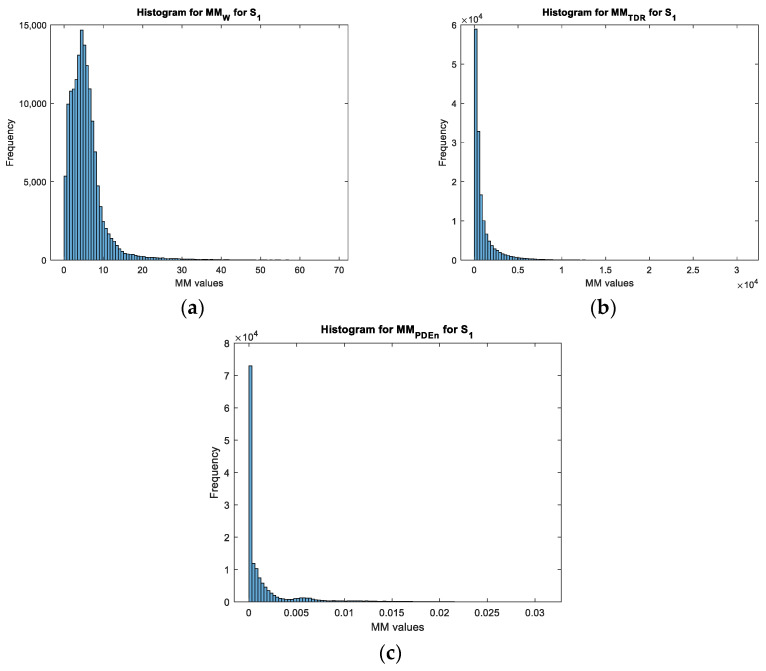
The histogram for the movement maps from S1 for the LSS UAV target: (**a**) The wavelet transform approach; (**b**) the TDR approach; and (**c**) the PDEn approach.

**Figure 12 sensors-23-01956-f012:**
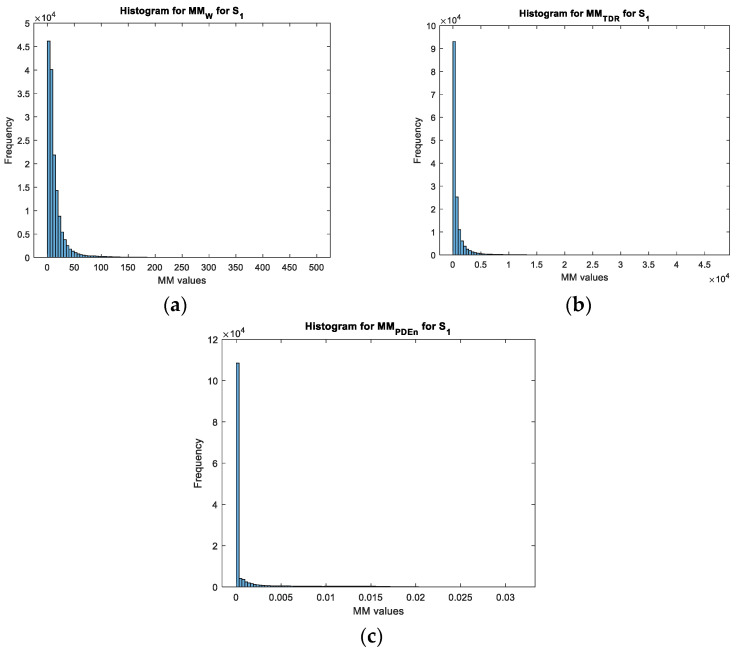
The histogram for the movement maps from S1 for the human target: (**a**) The wavelet transform approach; (**b**) the TDR approach; and (**c**) the PDEn approach.

**Figure 13 sensors-23-01956-f013:**
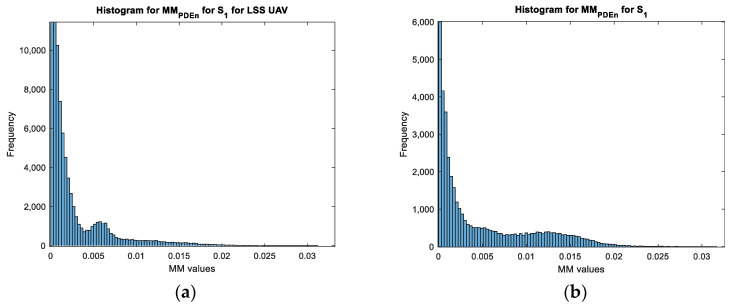
Zoom of the histograms from Figure 11c and Figure 12c: (**a**) LSS UAV movement; (**b**) human movement.

## Data Availability

The data presented in this study are available upon request from the corresponding author.

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
