# Peer review of "UWB Sensing for UAV and Human Comparative Movement Characterization"

_sensors, 2023, doi:10.3390/s23041956_

Round 1

Reviewer 1 Report

In this manuscript, the author proposed an approach to differentiate the movement of a drone and a human in an indoor environment, which is based on the combination of UWB sensing and advanced signal processing algorithms, i.e., TDR RPA or PDEn. The method is interesting, and the results are supported by the experimental demonstration. But I still have several questions as below:

Major comments:

1) The indoor measurements for drone and human are done separately or simultaneously? In this UWB frequency range, human would reflect most of the power and may block the signature of the UAV. Some discussions or more demonstrations may be added. 

2) Can this method be generalized with more UWB sensors, like 3 or 4? If yes,  will the crosstalk between those sensors generate any effects on the measurement? Some explanations could be added.

3) Some discussion about the algorithm is similar as in Ref [29], for example, about the TDR RPA. Maybe some explanations can be omitted. 

Minor comment:

1) Some labels in Fig. 1 and 2 are too small to be read. 

Author Response

First of all, we would like to thank the Reviewer for his relevant comments and suggestions, which allowed us to reconsider some aspects of our work and improve the paper content. We sincerely appreciate the time and effort you spent to review our paper.

We have provided point by point responses to the Reviewer’s comments. Your feedback is in italic font and red color, while our corresponding responses are in plain font and blue color. The paragraphs of the revised paper, which are modified according to the Reviewer’s comments, are highlighted in green.

In the revised paper, all the changes we have made are written in blue.

Reviewer 2 Report

I think the paper presents an interesting insight into the problem of drone detection in presence of humans and offers 3 similar methods of data processing all using data from ultra wideband detectors. I generally support this publication and think the problem is of importance and the methods are of potential significance.

However, I found a number of substantial shortcomings with the paper:

- It is not entirely clear what is new in this publication without referring to the cited literature, it would be good to highlight this.

- The logical structure of the paper and explanations need significant improvement. While the language is generally ok, the clarity must be improved.

- The comparative analysis of the methods and conclusions drawn are very questionable to me. A metric similar to signal-to-noise ratio or a blind analysis of some kind should be used to compare. In general, the data have been processed, but not analysed. If that is outside the scope of the paper, the scope needs to be defined. Although I think that the paper would be incomplete without a proper analysis and comparison of the described methods.

- It is not clear whether using 2 sensors one gains any advantage in terms of drone detection capability, the vertical motion is mentioned, but does not seem to figure in the measured data.

- Clearly, the disturbance introduced by humans in near range exceeds that of a relatively small drone, but there is also the factor of distance to the sensor. Again, either the scope needs to be defined, or multiple scenarios in terms of relative distances and direction of motion, including combined x/y/(z for the drone), need to be considered.

- Punctuation around equations is mostly wrong or missing.

- Equations are (unnecessarily) referenced before they are introduced.

I also have a number of more detailed comments to the text in its current form:

94 - define scan lines, what is scanned?

eq (3) - why i ranges from 1 to 399? what defines the nr of scan lines?

97 - what "its" refers to?

eq (5) - what does the index "w" stand for?

102-103 - text unclear

eq (6) - clarify indeces

118 - what are the chosen m and d?

122 - abbreviation RPA not explicitly defined

2.2 - needs a logic check

128-129 - should say how sudden changes are detected, and what "sudden" means

eq (10) - sum is over j?

\sigma - essentially defined twice

141-144 - not clear how the ellipce is found as without additional constraints miltiple ellipses could be constructed, which affects the resulting \sigma

162-163 - vs 168-169 - using two sensors because of dimensionality or providing a reference?

167-168 - movement of what?

170 - unclear subsentence

176 - could there be interference with the UWB detector from Bluetooth connection?

180 - which scenario?

183-184 - unclear, are "configurations" connected to the previous text?

Author Response

First of all, we would like to thank the Reviewer for his relevant comments and suggestions, which allowed us to reconsider some aspects of our work and improve the paper content. We sincerely appreciate the time and effort you spent to review our paper.

We have provided point by point responses to the Reviewer’s comments. Your feedback is in italic font and red color, while our corresponding responses are in plain font and blue color. The paragraphs of the revised paper, which are modified according to the Reviewer’s comments, are highlighted in green.

In the revised paper, all the changes are written in blue.

Round 2

Reviewer 1 Report

In this revised manuscript, the author has resolved all my questions, and thus, I don't have further concerns.